# Seed Germination Characteristics of a Critically Endangered Evergreen Oak—*Quercus marlipoensis* (Fagaceae) and Their Conservation Implications

**Luting Liu** [1,2,3] , **Yu Tu** [1], **Qiansheng Li** [4] **and Min Deng** [1,2,3,*]

1 Yunnan Key Laboratory of Plant Reproductive Adaptation and Evolutionary Ecology, Institute of Biodiversity, School of Ecology and Environmental Science, Yunnan University, Kunming 650500, China; liuluting0115@gmail.com (L.L.); orchid-txx@mail.ynu.edu.cn (Y.T.)

2 Yunnan Key Laboratory for Integrative Conservation of Plant Species with Extremely Small Populations, Kunming Institute of Botany, Chinese Academy of Sciences, Kunming 650201, China

3 The Key Laboratory of Rare and Endangered Forest Plants of National Forestry and Grassland Administration & The Key Laboratory for Silviculture and Forest Resources Development of Yunnan Province, Yunnan Academy of Forestry and Grassland, Kunming 650201, China

4 Southeastern Center for Conservation, Atlanta Botanical Garden, Atlanta, GA 30309, USA; qli@atlantabg.org

\* Correspondence: dengmin@ynu.edu.cn; Tel.: +86-136-6172-0514

**Abstract:** Seed germination is among the most crucial and vulnerable stages in plant life cycles. *Quercus marlipoensis* is a critically endangered sclerophyllous oak. Only one population has ever been found worldwide in the tropical montane cloud forests of southeastern Yunnan, China, and it has shown difficulties with regeneration. However, its seed biological traits and key restrictive germination factors remain unknown. We investigate the impacts of scarification, temperature, and water potential on the seed germination of *Q. marlipoensis*. Results show that the seeds show typical epicotyl dormancy. The seed germination increased when removing part or all of the pericarp and part of the cotyledon (one-third and two-thirds). The seeds can germinate at 5 to 30 °C, but the highest $T_{50}$ was achieved at 25 °C. When the water potential decreased from 0 to −1.0 MPa, the germination rate decreased but the germination time increased. *Q. marlipoensis* seeds are typically recalcitrant and highly sensitive to moisture loss, but the species can tolerate animal predation and low germination temperatures. The more frequent climatic extremes and droughts in the Indo-China region will severely degrade its natural habitats. Therefore, ex situ conservation to preserve its germplasm and introduce seedlings into a suitable habitat are essential for its conservation management.

**Keywords:** *Quercus* section *Ilex*; tropical montane cloud forests; seed germination; acorn predation; water potential





## 1. Introduction

Tropical montane cloud forests (TMCFs) are unique narrow altitudinal belts in tropical highlands [1,2] between 1000 and 3000 m above sea level [3]. Their microclimate is characterized by persistent, frequent, or seasonal cloud cover at the vegetation level, with fog and mist surrounding the canopy [4]. The structure and function of TMCFs are highly vulnerable to climate change, particularly temperature increase [5], which reduces mist and causes the extinction of cloud forests on mountain peaks [6].

*Quercus marlipoensis* Hu (section *Ilex*) is an endemic evergreen sclerophyllous oak distributed in Malipo County, Yunnan Province, China [7]. This species was first reported by Hu (1951) [8] based on the voucher specimen collected in 1940 by C. W. Wang. Since then, no new specimens nor occurrence data have been reported for decades, even after three national plant inventory surveys were conducted from 1990 to 2010. However, our field expedition in 2012 rediscovered this species in the core protection area of Lao-Jun-Shan National Nature Reserve, where the specimen originated [9]. Still, no other population

has been discovered in the past ten years after extensive field surveys. The population showed significant signs of regeneration difficulties, as seedlings or saplings were relatively rare. The IUCN Red List classifies this species as Critically Endangered (CR) [10]. The newly published book *List of Yunnan Protected Plant Species with Extremely Small Populations* (PSESP) [11] also included this species. However, the reasons for the rarity of *Q. marlipoensis* and the possible factors threatening its regeneration are still unknown. Understanding the important physiological and ecological characteristics of *Q. marlipoensis* is crucial for identifying its threatening factors and conservation strategies.

Poor regeneration is one of the factors that lead to the decline of threatened species [12]. Knowledge of the seed germination characteristics of PSESPs is important for the conservation of threatened species [12,13]. Biotic and abiotic stress can profoundly impact the seed germination and seedling establishment process. Cotyledons of oak seeds rich in starch and other nutrients are a dietary staple of rodents and insects and the foundation of the food chain in a forestry ecosystem [14]. Meanwhile, the energy reserves in cotyledons are also an important source supporting seedling development [15–17]. If animals in the ecosystem intensively consume oak seeds, seed mortality will dramatically increase, preventing tree populations from regenerating [18–21]. Mechanical scarification is a practice of simulating animals preying on seeds, and different treatments (pericarp or partial cotyledon removal) resemble the degree to which the seeds are eaten by animals [22,23]. Oak seeds show very different levels of sensitivity to animal predation among species. Previous mechanical scarification studies showed that losing a small part of the cotyledons in acorns of various oak species only has a minor impact on seedling establishment [24–26] and may even increase the germination ratio [26,27]. Under loss of cotyledons, the responses of seeds are quite different among oak species, e.g., the hypocotyl and radicle mortality dramatically increased in *Q. variabilis* [28], but *Q. robur* seeds can successfully germinate and establish seedlings even when they have suffered from severe cotyledon reductions of up to two-thirds [27]. Evaluating seed tolerance to animal predation is an essential step to untangling the factors limiting the regeneration of *Q. marlipoensis*.

Temperature and moisture (water potential) are the key environmental factors impacting seed germination [12,29]. Oak seeds are mostly recalcitrant, with a moisture content of 30%–55%, and seed moisture loss significantly decreases the germination ratio [30–34]. The majority of seeds exhibit rapid germination and continuously maintain high activity without dormancy characteristics post-maturation [35,36]; however, a small percentage of oak seeds may undergo dormancy [37], influenced by the effects of growth regulators, pericarp restrictions, or a combination of both [38,39], breaking dormancy in oak seeds typically requires stratification or pericarp abrasion [40]. As a widespread Northern Hemisphere woody clade, oaks show high ecological divergence. Their habitats range from humid temperate and tropical forests to semi-arid deserts [41,42]. Correspondingly, the best germination temperatures of oaks have typically been confirmed to match their habitats' climatic features [43], ranging from 5 °C to 35 °C [43,44]. Although oak habitats are quite different, their seeds generally show a degree of cold tolerance, as most of them can germinate at a low base temperature ($T_b$) of about 5 °C [29,43,45]. Overall, the currently published seed physiological research on oaks is mainly on widespread oaks. The germination traits of endangered oaks remain largely unknown.

Considering the low regeneration capacity of *Q. marlipoensis* in its natural habitat, it is crucial to reveal the key factors restricting its seed germination and seedling establishment to provide conservation guidelines for this extremely endangered oak species. This study was undertaken to reveal the effects of scarification, temperature, and water potential on the seed germination of *Q. marlipoensis*. This study provides crucial information on this critically endangered oak in order to provide guidelines on its in situ and ex situ conservation.

## 2. Materials and Methods

### 2.1. Study Area and Acorn Collection

The habitat of *Quercus marlipoensis* is located at Xia-Jin-Chan village of the Lao-Jun-Shan National Nature Reserve (23.40859° N, 104.70901° E) of southeast Yunnan Province, China. It is found in TMCFs at the middle-elevation (1700–1900 m above sea level) karst areas characterized by high humidity and warm temperatures (Figure 1). The region is under the Indian summer monsoon regime with prominent warm rainy summers and mild, dry winters. The annual highest and lowest temperatures are 28.10 °C and 6.50 °C, respectively. The annual accumulated temperature greater than 10 °C is about 4500 °C to 7500 °C, and the annual average precipitation is 1318 mm [6]. In the region, we visited residents and conducted surveys of all potential forests to search for additional populations of *Q. marlipoensis*. However, only 64 mature and 4 young trees were scattered throughout the forested area.

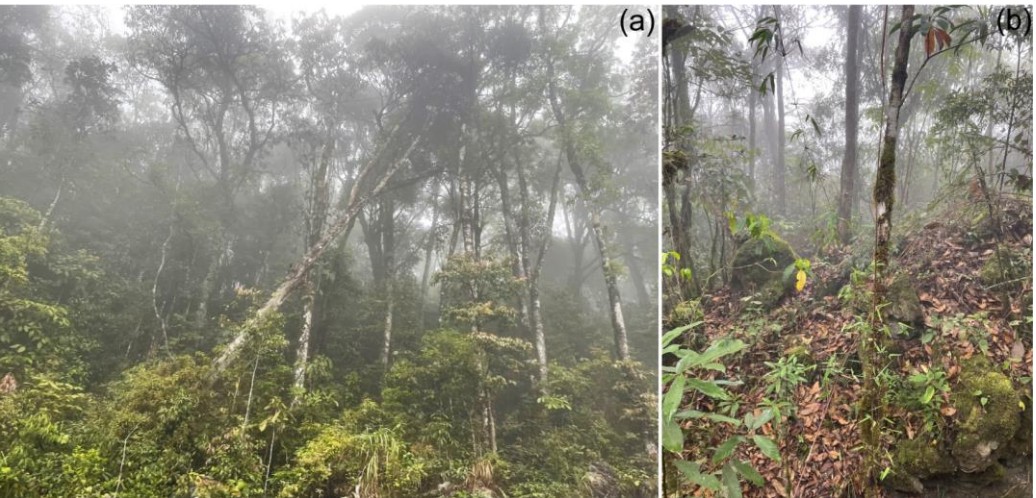

**Figure 1.** The natural habitats of *Q. marlipoensis* in Xia-Jin-Chang village, Lao-Jun-Shan National Nature Reserve, Malipo County, southeastern Yunnan, China. (**a**) The landscape of tropical montane cloud forests (TMCF); (**b**) A close view of the community showing the limestone rocky soil outcrops.

Acorns of *Q. marlipoensis* were collected randomly in late October 2021. At least 4 kg of seeds were collected from different *Q. marlipoensis* adult individuals. The collected seeds were then randomly distributed into three groups for morphological measurements, dry and wet weight assessments, and seed germination experiments, respectively. Before commencing the experiments, thorough visual inspections were conducted on the acorns. Any acorns displaying abnormalities or noticeable defects were removed from the selection.

### 2.2. Morphological Traits of the Acorn and Seed

The length (L), cupule length (CL), width (W), and seed scar diameter (CSD) of 30 random fresh acorns were measured with a vernier caliper. The morphological variation in acorns and the longitudinal section of the seeds were captured in images using a stereo-microscope (Leica S9I, Am Leitz-Park, Wetzlar, Germany).

The acorns were divided into three batches (*n* = 100) to determine their fresh weight and then were dried at 103 ± 2 °C in an oven for 17 h to obtain the dry weight following a previously described protocol [46] (International Seed Testing Association 1999). The initial moisture content of acorns was calculated as (fresh weight − dry weight)/fresh weight × 100% [47].

### 2.3. Seed Pretreatment and Germination Test

After removing the cupule, the acorns were soaked in deionized water for two hours. The floating acorns were discarded. The sunken acorns were selected and surface-disinfected

by soaking them in 1% sodium hypochlorite solution for 0.5 h, then rinsing them three times using deionized water [47] before the subsequent germination tests. All treatments in the three experiments were carried out with three replicates, and each replicate included 20 acorns. Plump acorns of similar size were selected and placed in covered plastic boxes, with two layers of filter paper on the bottom of the box as the germination substrate. Seeds germinated under conditions of 25 °C, with 16 h of light followed by 8 h of darkness; germination humidity was near 100 percent [48].

For the scarification experiment, five treatments were performed on the acorns: intact seeds with pericarp (CK), removal of cup scar (RS), removal of the pericarp (RP), cutting off 1/3 of the cotyledon (TC), and removal of the pericarp and cutting off 2/3 of the cotyledon (OC).

Plump acorns of similar size were selected and placed in covered plastic boxes. The boxes were placed in growth chambers spanning a set temperature gradient (5 °C, 10 °C, 15 °C, 20 °C, 25 °C, and 30 °C).

To evaluate the effects of water stress on seed germination, different concentrations of PEG-6000 solutions (Yatai United Chemical Co., Ltd., Wuxi, China) with corresponding water potentials of 0, −0.2, −0.4, −0.6, −0.8, and −1.0 M pa [49] were added to the bottom of the plastic boxes along the edge of the filter paper until the filter paper was completely saturated. The filter papers and water/PEG solutions were replaced every other day.

*2.4. Calculation of Seed Germination Indices*

Seeds were considered germinated when the radicle emerged more than 2.0 mm [43]. The germinated seeds were counted and removed every other day. Counting was ended when either all of the seeds had grown or no germination was observed for 30 consecutive days [46].

The statistical analysis and visualization of germination data were performed using the 'Germinationmetrics' package of R version 4.1.2 [50]. The indices for evaluating seed vigor we selected were germination percentage, the initial germination time, the last germination time, time for 50% of seeds to germinate ($T_{50}$), germination value, and germination index [51–53].

The *x*-intercept method was used to determine the base temperature ($T_b$) and base water potential ($\psi_b$). Germination percentage was calculated as the inverse of $T_{50}$, and linear regressions were conducted to determination its relationship with temperature and water potential, respectively. The *x*-intercept of the linear equation was calculated as $T_b$ or $\psi_b$ [54,55].

*2.5. Statistical Analyses*

Analysis of variance (ANOVA) was used to assess the effects of different scarification methods, temperature, and water potential on seed germination. Germination percentages were analyzed after arcsine transformation because of the non-normal distribution of the data. The differences between means were tested using Tukey's HSD test at a $p \leq 0.05$ threshold. All statistical analyses were conducted with R version 4.1.2.

## 3. Results

*3.1. Morphological Characteristics of the Acorn and Seed Germination*

The average fresh weight of each acorn was 3.12 g (derived from 100-seed weight). The moisture content of acorns was 55.34% ± 1.56%. The acorn is ellipsoidal with a cup-shaped cupule. The purplish-red triangular bracteoles were imbricated on the outside of the cupule and more prominent at the top (Figure 2a). The mean length and width of the acorn were 1.77 ± 0.17 cm and 1.39 ± 0.08 cm, respectively. The length of the cupule was 1.01–1.45 cm (average 1.22 cm), and the cup scar diameter (also known as seed scar diameter) was 0.45–0.67 cm (average 0.56 cm). The seed cotyledon is creamy white or light yellow (Figure 2c). The radicle penetrates the seed coat at or beside the remains of the

style when germinating (Figure 2d,e). The epicotyl did not elongate until 12 weeks after germination when the radicle was 7.00–9.00 cm long (Figure 2f).

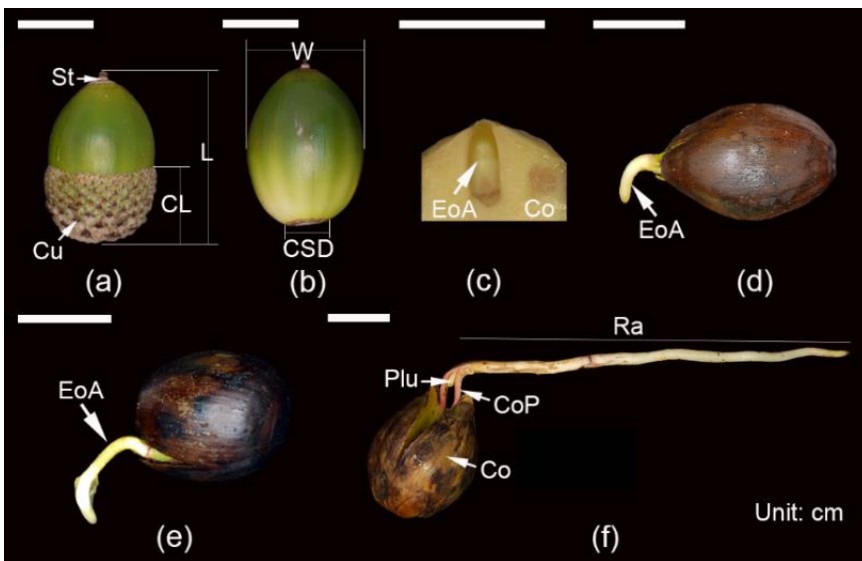

**Figure 2.** Acorn and seed germination morphology of *Q. marlipoensis*. Scale bar = 1.0 cm. (**a,b**) Mature acorn with cupule (**a**) and without cupule (**b**); CL indicates the cupule length measurement; L and W represent the acorn length and width, respectively; CSD indicates the cupule scar. (**c**) Embryo position in the seed. (**d,e**) Radicle emergence at day 2 of germination, showing that embryo position varies from near the remains of the style (**d**) to beside the remains of the style (**e**). (**f**) Seed after two weeks of germination, an elongated radicle and emergence of the plumule. St: remains of the style; Cu: cupule; Co: cotyledon; EoA: embryo axis; CoP: cotyledon petiole; Plu: plumule; Ra: radicle.

*3.2. Seed Germination Traits*

3.2.1. Scarification and Germination

Seeds subjected to pericarp removal and/or partial cotyledon removal showed 90.00% to 100.00% germination percentages, while only 73.33% germinated in the control group. Pericarp removal also significantly accelerated the speed of germination (Table S1, RP_$T_{50}$ = 4.70 ± 0.55 day, CK_$T_{50}$ = 19.88 ± 1.63 day). The initial germination time showed no significant difference among the five treatments (Table S1, $p > 0.05$), but the last germination time of the acorns without pericarp tissue was earlier than that of the control and RS groups (Table S1, Figure 3a). The significant decrease in $T_{50}$ and increase in both the germination value and the germination index (Figure 3b–d) indicated that the intact pericarp impeded germination. However, cup scar removal decreased $T_{50}$ and increased the germination index less significantly than removal of the whole pericarp or cutting off part of the cotyledon (RP, TC, OC).

3.2.2. Temperature and Germination

Seeds at all six of the tested temperatures (from 5 °C to 30 °C) had high cumulative germination percentages ranging from 70.00% to 100.00% (Table S2, Figure 4a). The initial and last germination times at 5 °C and 10 °C were significantly later than those at 15–30 °C (Table S2 and Figure 4a, $p < 0.05$). The time to complete germination was shortened significantly above 15 °C. $T_{50}$ increased significantly at 5–15 °C and remained low at 20–30 °C (Table S2 and Figure 4b). The germination value and germination index increased with temperature (Figure 4c,d).

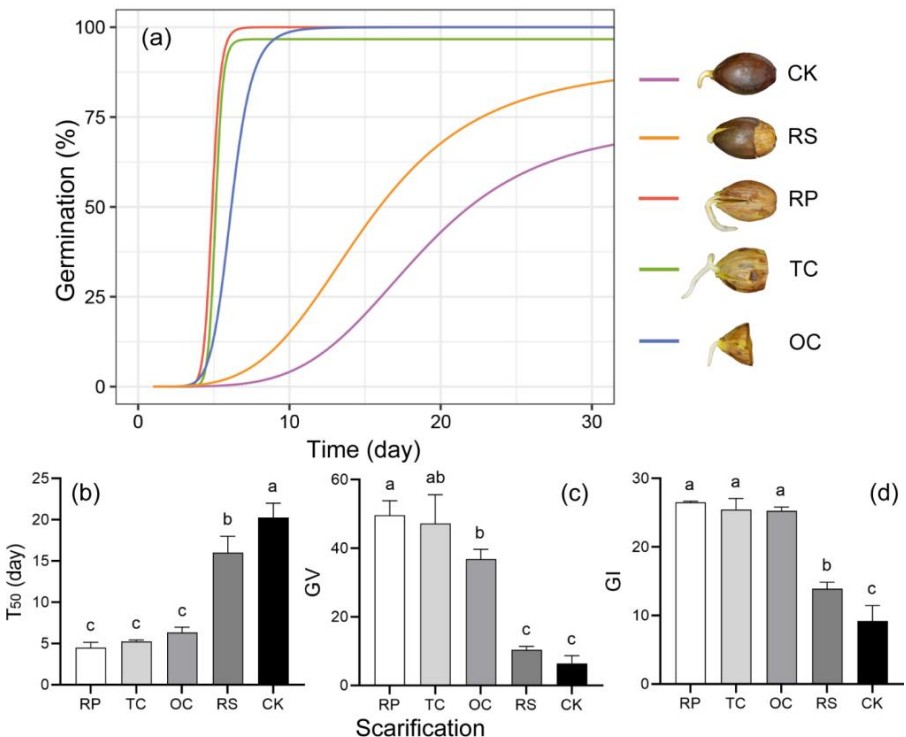

**Figure 3.** Seed germination patterns of *Q. marlipoensis* under different pericarp removal and cotyledon cut-off treatments. (**a**) Fitted germination curves, where CK represents seeds with pericarp, RS represents removal of cup scar, RP removal of the pericarp, TC and OC respectively represent seeds with pericarp removal and cutting off 1/3 of cotyledons and 2/3 of the cotyledon; (**b**) $T_{50}$; (**c**) germination value (GV); (**d**) germination index (GI). Means followed by the same letter are not significantly different at $p = 0.05$ (Tukey HSD test).

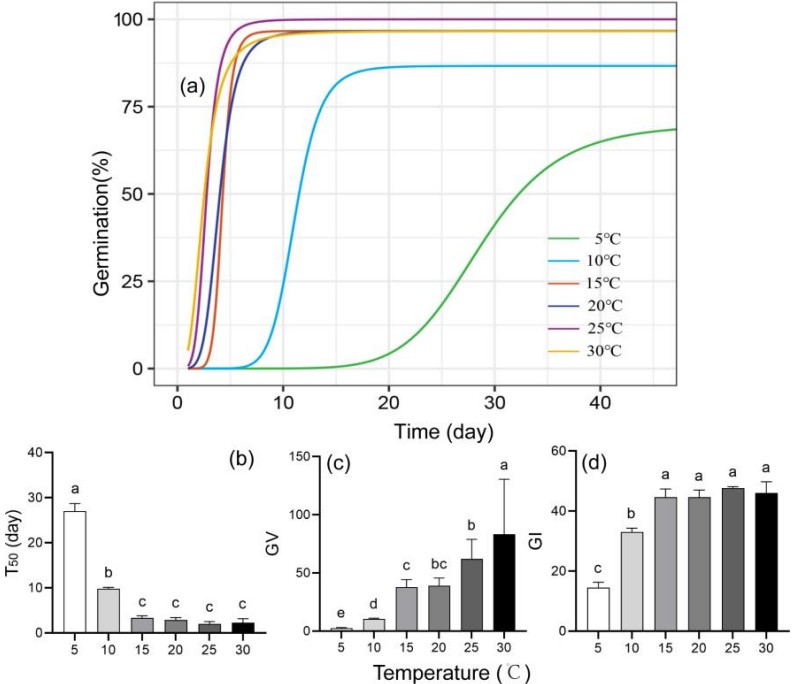

**Figure 4.** Seed germination patterns of *Q. marlipoensis* at different temperatures. (**a**) Germination curves; (**b**) $T_{50}$; (**c**) germination value (GV); (**d**) germination index (GI). Means followed by the same letter are not significantly different at $p = 0.05$ (Tukey HSD test).

GR increased linearly with temperature, reaching 0.57 days$^{-1}$ at 30 °C, though the final germination percentage did not increase from 15 °C to 30 °C. The estimated $T_b$ was 3.60 °C (Figure 5).

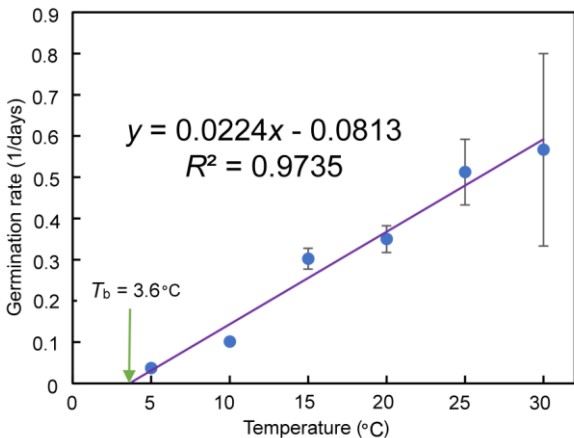

**Figure 5.** Effect of temperature ($x$) on the germination rate (GR, $y$) of *Q. marlipoensis*. The germination base temperature ($T_b$) was calculated as the x-intercept of the linear regression ($y = 0.0224x - 0.0813$, $R^2 = 0.9735$). GR was calculated as the inverse of $T_{50}$.

### 3.2.3. Water Potential and Germination

Water potential remarkably affected germination percentage, $T_{50}$, germination value, and germination index (Table S3, Figure 6). Lower water potential (i.e., more negative values) reduced the final germination percentage. The seeds subjected to the control treatment (0 MPa) started to germinate within 3 days (Table S3), while those under water stress took 2 to 4 weeks to start germination (Figure 6a). $T_{50}$ extended from 3.6 days to 34 days across water potential values from −0.2 MPa to −1 MPa (Figure 6b). In other words, germination significantly slowed as water potential decreased. Both germination value and germination index decreased as water potential decreased (Figure 6c,d).

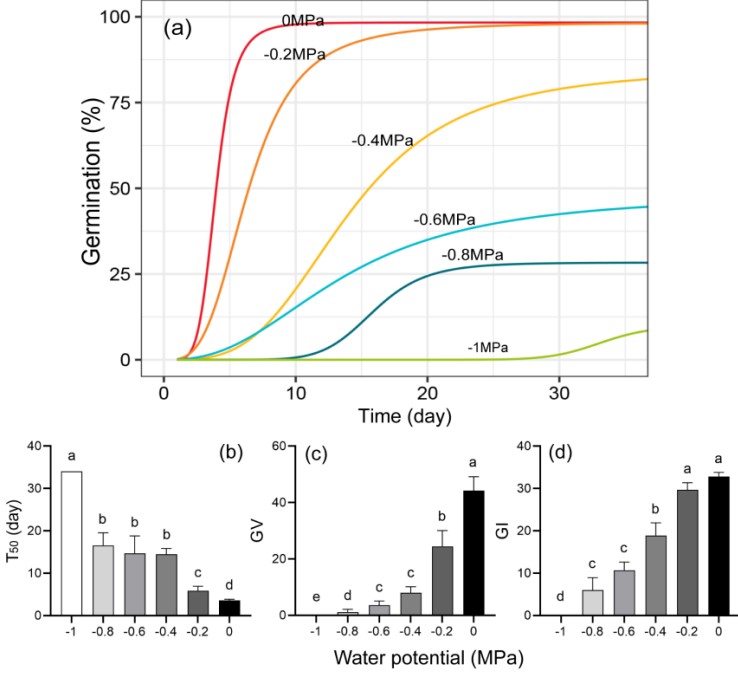

**Figure 6.** Seed germination pattern of *Q. marlipoensis* under different water potentials. (**a**) Fitted germination curves; (**b**) $T_{50}$; (**c**) germination value (GV); (**d**) germination index (GI). Means followed by the same letter are not significantly different at $p = 0.05$ (Tukey HSD test).

The final germination percentages decreased linearly with water potential between $-0.2$ and $-1.0$ MPa (Figure 7). Almost all the seeds germinated under 0 MPa and $-0.2$ MPa. The germination percentage also decreased linearly with water potential (Figure 8). The base water potential ($\psi_b$) was $-0.9$ MPa according to the *x*-intercept of the linear regression.

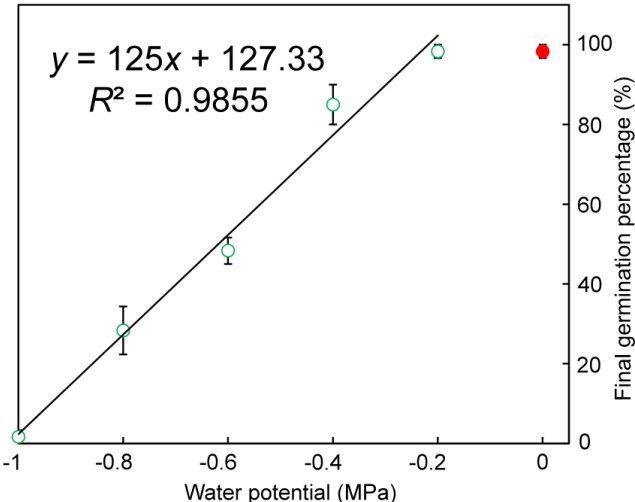

**Figure 7.** Effect of water potential ($\psi$) on the final germination percentage (%) of *Q. marliponesis*. The linear equation ($y = 125x + 127.33$, $R^2 = 0.9855$) fit the data well when data points for 0 MPa treatment (which had the same germination percentage as $-0.2$ MPa treatment) were omitted.

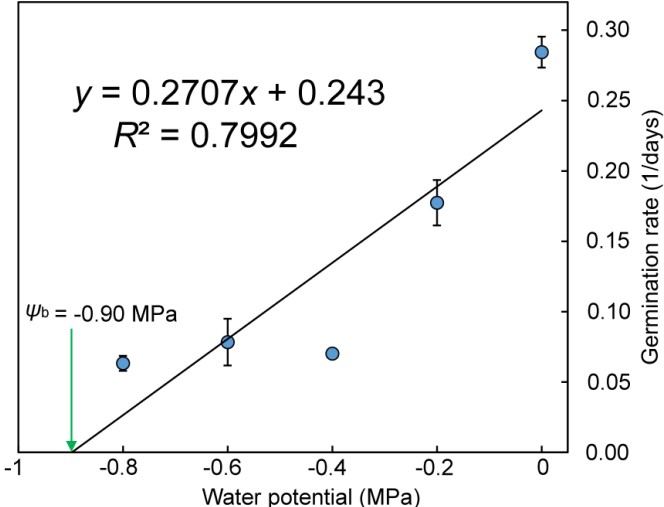

**Figure 8.** Effect of water potential ($\psi$) on the germination rate (1/days) of *Q. marliponesis*. The linear equation ($y = 0.2707x + 0.33$, $R^2 = 0.7992$) was fitted to the data, and the *x* intercept was calculated to estimate the base water potential ($\psi_b$).

## 4. Discussion

### 4.1. Impacts of Pericarp and Cotyledon Removal on the Germination of Quercus marlipoensis

Animal predation on the pericarp and cotyledons of acorns may accelerate seed germination and increase the seed germination percentage [26,56]. Our results showed that the complete removal of the pericarp and partially cutting off the cotyledons significantly improved the final germination percentage and shortened the germination time of *Q. marlipoensis*. The removal of the cup scar enhanced the germination percentage and shortened the germination time, but the influence was not as great as the complete pericarp removal treatment, and this result is consistent with findings from other studies on oak seeds [23,27]. Pericarp removal improves water absorption and gas exchange, boosting

the germination ratio [57]. The observed improvement in germination could be explained by the removal of both the mechanical barrier and inhibitory substances in the pericarp and/or cotyledon [48,58,59].

Although *Q. marlipoensis* is critically endangered, its seeds can individually tolerate a medium-high level of animal predation. The influence of cotyledon loss by animal consumption on oak seed germination and seedling survival varies by species. For example, a loss of more than 50% of the cotyledon tissue has a significant negative effect on seed germination and seedling growth in *Q. variabilis* [28,60] as well as *Q. prinus* and *Q. velutina* [61]. Our results showed that loss of two-thirds of the cotyledon tissue had no significant Impact on the germination of *Q. marlipoensis*, suggesting that *Q. marlipoensis* only needs a little nutrition from its cotyledons to ensure its germination and that it has good tolerance to animal predation. Likewise, similar results were obtained in *Q. robur* [27] and *Q. mongolica* [62]. During the coevolution between oaks and seed predators, oaks tend to develop large acorns or extra nutrition to ensure germination, even under high predation pressure [63,64].

### 4.2. Effect of Temperature on Seed Germination and the Base Temperature

Warm environmental conditions are generally suitable for seed germination, but some seeds can also tolerate low temperatures. The estimated base temperature for *Q. marlipoensis* seed germination is 3.60 °C. Its seeds can germinate at high percentages at temperatures above 5 °C, and germination sped up once the temperature increased. Temperatures from 15 °C to 30 °C promoted similarly high germination percentages and speeds. Similar results were found among different source populations of *Q. robur* from England and the Netherlands [65]. The threshold values are linked to the species' geo-climatic origins [29]. Oak species from subalpine, subtropical, and temperate regions have been shown to have a similar low $T_b$, but different germination percentages [43]. In contrast, oaks in the section *Cyclobalanopsis* from tropical and subtropical Asia cannot germinate well at 15 °C [66]. The difference in the seed germination temperature of oak species from different biomes indicates that local adaptation indeed plays a role in the response to diverse environmental factors. Likewise, our study showed that seeds of *Q. marlipoensis* have a germination percentage of 70.00% at 5 °C, with $T_b$ at 3.60 °C, which is much higher than that of evergreen oaks from subtropical lowlands. The ability to germinate at low temperatures suggests that *Q. marlipoensis* can cope with cool temperatures during the germination period under sufficient water content conditions, which is compatible with the microhabitat of intermediate-elevation TMCFs.

### 4.3. Effect of Water Potential on Seed Germination

*Quercus marlipoensis* is a recalcitrant-seeded species that is highly sensitive to desiccation, but its seeds can germinate quickly in moist conditions. Our results showed that *Q. marlipoensis* acorns are unable to tolerate osmotic potentials lower than −0.2 MPa. Therefore, its seeds can only successfully germinate and establish seedlings in humid soils. This result is similar to those found in other endangered *Quercus* spp. (e.g., *Q. insignis*, *Q. sartorii*, and *Q. xalapensis*) in TMCFs in Mexico, which can only grow vigorously in humid understory forests [67,68]. Rapid germination of various species of *Quercus* could be a strategy to reduce desiccation-induced mortality [43] and avoid predation [56].

Nevertheless, oaks in different habitats show different levels of resistance to water loss during germination. For example, seed germination percentage and germination percentage of *Q. leucotrichophora* from the dry, hot valleys of the central Himalayas were inhibited under a −0.9 MPa water potential, but the species still had a 20% final germination percentage [69]. Tropical trees growing in humid regions require a high base water potential for germination [29,57], and their tolerance to drought in the subtropical humid forests is much lower than that of seeds from tree species in arid environments [70,71].

### 4.4. Conservation Implications

The seed germination process of *Q. marlipoensis* is highly sensitive to humidity loss in the soil (i.e., low water potentials of −0.8 MPa and −1 MPa), and it can germinate rapidly under high water potentials. This characteristic is compatible with the unique cool and high-humidity microclimate of TMCFs. It is worth noting that *Q. marlipoensis* grows in limestone areas characterized by higher soil water evaporation rates and lower soil moisture than those of other soil types [72–74]. Such soil conditions could make *Q. marlipoensis* more vulnerable to drought stress.

Southwest China is under the Indian summer monsoon regime, with a prominent dry season beginning in September and lasting until the subsequent April [75]. The unique topography and microhabitats of TMCFs ensure the high humidity of these forests. However, the increasing length of the seasonal dry period coupled with more frequent climate extremes, e.g., drought and extreme heat events, throughout Southwest China and Indo-China as a result of global climate change profoundly impacts the local biota [76–78], which also degrades the TMCF habitat and profoundly impacts its natural seed germination and seedling establishment processes. However, oak acorns are not efficient propagules for long-distance dispersal that can track suitable habitats over time. Therefore, in situ conservation which first introduce the seeds to nurseries/botanical gardens and then reintroduce large seedlings back into suitable habitats with anticipated long-term stability will be an effective solution to help maintain the population size of *Q. marlipoensis*. Moreover, a growing body of species distribution dynamics studies have predicted a general trend of northward range expansion pattern of Asian evergreen broadleaved forest tree species [79–81]. Nevertheless, the other physiological characteristics of *Q. marlipoensis* are still unknown. Future studies coupling species distribution modeling and physiological analysis in *Q. marlipoensis* will provide sufficient guidance to the conservation of this endangered oak in the context of global climate change.

### 5. Conclusions

*Quercus marlipoensis* is a critically endangered tree restrictively distributed within the TMCFs of southeastern Yunnan, China. Its seeds are recalcitrant, with typical epicotyl dormancy; they germinate quickly once mature, but epicotyl development is significantly delayed. The acorns of *Q. marlipoensis* can each tolerate medium-high animal predation, and its seeds are viable at low germination temperatures, with a relatively high germination percentage. Its seed germination process is highly sensitive to humidity loss. The germination process of *Q. marlipoensis* is compatible with the microenvironments of TMCFs. But it only grows in limestone areas, and this soil type is characterized by higher soil water evaporation and low soil moisture when under environmental drought conditions. The increasing length of the seasonal dry period and more frequent climate extremes in Southwest China and Indo-China in recent years has led to habitat degradation, resulting in difficulties with seed germination and seedling establishment of *Q. marlipoensis* in its natural habitat. Therefore, manual ex situ germplasm conservation aiming to introduce seeds to nurseries and botanical gardens for propagation and subsequently reintroduce large seedings to suitable habitats should be carried out to conserve this extremely endangered oak species.

**Supplementary Materials:** The following supporting information can be downloaded at: https://www.mdpi.com/article/10.3390/f15020235/s1, Table S1: Seed germination indices of *Quercus marlipoensis* under different pericarp removal and cotyledon cut-off treatments of, Table S2: Seed germination indices of *Quercus marlipoensis* at different temperatures, Table S3: Seed germination indices of *Quercus marlipoensis* under different water potentials.

**Author Contributions:** M.D. conceived the idea. L.L. and Y.T. performed the experiments, and L.L. collected the data. L.L. and Q.L. analyzed the data. L.L., M.D. and Q.L. wrote the manuscript. All authors have read and agreed to the published version of the manuscript.

**Funding:** The study was supported by the National Natural Science Foundation of China (grant. no. 31972858), the International Oak Society (IOS) Oak Conservation and Research Fund, the Fund of Yunnan Key Laboratory for Integrative Conservation of Plant Species with Extremely Small Populations (grant. no. PSESP2021), and the Project of the Yunnan Academy of Forestry and Grassland (grant. no. KFJJ21-05), the project of the Southeast Asia Biodiversity Research Institute, Chinese Academy of Sciences (grant. no. Y4ZK111B01), and National Key Research and Development Program of China 2023YFF1305000.

**Data Availability Statement:** The data presented in this study are available on request from the corresponding author. The data are not publicly available due to privacy.

**Acknowledgments:** We are grateful to the Malipo Lao-Shan Provincial Nature Reserve of Yunnan Province, the Asian Elephant Yunnan Field Scientific Observation and Research Station, the Yunnan Asian Elephant Field Scientific Observation and Research Station of the Ministry of Education, and the Baima Snow Mountain Complex Ecosystem Vertical Transect Field Observation for their help with the fieldwork and seed collecting. We thank Lin Lin, Yuxin Ma, and Chaoying Yang for their help with data analyses.

**Conflicts of Interest:** The authors declare no conflicts of interest.

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
