# Peer review of "Seed Germination Characteristics of a Critically Endangered Evergreen Oak—Quercus marlipoensis (Fagaceae) and Their Conservation Implications"

_forests, doi:10.3390/f15020235_

Round 1

Reviewer 1 Report

Comments and Suggestions for Authors

The present document evaluates the fermination requirements of Quercus marlipoensis.

The abstract reflects the results and conclusions obtained.

The introduction provides a large overview well referenced and presents clearly the research question.

The material and method is well described.

The results are presented in a scientific way, figures are of really good quality.

The discussion is also well referenced and complete.

The conclusion matches with the results.

Only minor three minor form issues were detected, please find the attached file the exact place.

Author Response

Dear Reviewer,

I would like to express my gratitude for your insightful feedback on my manuscript. Your suggestions have been invaluable in improving the quality of the paper.

I have addressed each of the comments and made corresponding revisions to the manuscript. Please see the attachment.

Thank you once again for your time and consideration.

Reviewer 2 Report

Comments and Suggestions for Authors

Dear Authors,

With considerabe interest I have read Your manuscript titled „Seed germination characteristics of a critically endangered evergreen oak—Quercus marlipoensis (Fagaceae) and its conservation implication” and I have found it generally well-written. Nevertheless, I have found some imperfections, which- in my opinion- should be improved or at least clarified before an eventual publication.

I have listed them below:

1. The choice of  Quercus marlipoensis for investigations should be better justified in chapter Introduction. 

2. I would like to encourage Authors to prepare separate chapter titled A study species containing  characteristics Quercus marlipoensis with information about range, habitat affiliation, lifespan, reproduction mode, dispersal.

3. In my opinion the characteristics of study area should be enlarged. The information about population od Q. marlipoensis (e.g. number of idividuals or area of population, distribution of individuals, age or size structure if it is possible) should be added. Text in lines 43-53 should be moved into section Study area. 

4. Material and methods-please describe more detaidly the way of collection of acors (if they were collected randomely or systematicaly, how many acorns were collected)

5. Please, improve the quality of Figures. In present form they are illegible.

Author Response

(The authors gave the same response as above.)

Reviewer 3 Report

Comments and Suggestions for Authors

The manuscript deals with the interesting and current issue of protection of genetic resources of plants in ex situ. The issue of protecting or restoring original populations of organisms, i.e. not only plants, is still at the forefront of the interests of not only the professional public. For this reason, I consider the submitted manuscript to be significant and stimulating. The manuscript is relatively well written, but I agree that it should have been supplemented or modified. In the introductory part, I somewhat miss the relationship between dormancy and post-dormancy on oak germination. This factor can also play a role. It would be appropriate for the authors to state this. The methodology states that part of the fruit was dried to determine the percentage of water, but the germination methodology is described in the next section. To an uninitiated reader, this could lead to the impression that these are the same group of seeds. It would therefore be appropriate to add that the seeds (fruits) were divided into two groups. From the text of the methodology, it is not clear whether the seeds germinated in the light or in the dark, as this factor is also important for some types of plants. The graphs showing the results are small and confusing, it is very difficult to find the values in the text. I recommend enlarging them or converting them into clearer tables. The discussion is descriptive in places, so I recommend reviewing it. Are all older literary sources necessary?

Author Response

(The authors gave the same response as above.)

Round 2

Reviewer 2 Report

Comments and Suggestions for Authors

Dear Authors,

I do not have any further remarks.